# Basic Guide to Multilayer Microfluidic Fabrication with Polyimide Tape and Diode Laser

**DOI:** 10.3390/mi14020324

**Published:** 2023-01-27

**Authors:** Thana Thaweeskulchai, Albert Schulte

**Affiliations:** School of Biomolecular Science and Engineering, Vidyasirimedhi Institute of Science and Technology (VISTEC), Wang Chan Valley, Rayong 21210, Thailand

**Keywords:** diode laser, polyimide, microfluidic, capillary, passively driven, multilayer

## Abstract

For normal operations, microfluidic devices typically require an external source of pressure to deliver fluid flow through the microchannel. This requirement limits their use for benchtop research activities in a controlled static environment. To exploit the full potential of the miniaturization and portability of microfluidic platforms, passively driven capillary microfluidic devices have been developed to completely remove the need for an external pressure source. Capillary microfluidics can be designed to perform complex tasks by designing individual components of the device. These components, such as the stop valve and trigger valve, operate through changes in microchannel dimensions and aspect ratios. A direct, maskless fabrication protocol that allows the precise fabrication of microchannels and other microfluidic components is introduced here. A diode laser and polyimide tape on a PMMA substrate are the only components needed to start fabrication. By varying the laser power used and the number of laser repetitions, various depths and widths of the microchannel can be quickly created to meet specific needs. As an example of a functional unit, a trigger valve was fabricated and tested, as proof of the validity of the fabrication protocol.

## 1. Introduction

Microfluidic devices have been a part of researchers’ toolkits for many years, serving various functions ranging from electrochemical sensing [1,2,3,4,5] to tissue engineering [6,7]. A conventional microfluidic experimental setup usually includes the microfluidic device and an external apparatus that supplies pressure to drive fluid flow within the device. The two devices together are designed to be operated together as an immobile experimental setup on a laboratory benchtop. The limited mobility of this setup fails to take full advantage of microfluidic devices that are miniaturized and portable analytical equipment, such as point-of-care (POC) devices. To address this shortcoming, a novel class of passively driven microfluidic devices that requires no external source of pressure for operation was created.

Capillary microfluidic devices rely on the intrinsic capillary pressure within the microchannel as the only source of power driving the fluid flow [8]. A previous study determined the specific microchannel parameters necessary for spontaneous capillary flow (SCF) to occur, namely, the cross-sectional area of the wetted perimeter, the free air–liquid interface or free perimeter, and the maximum contact angle [9]. Based on these parameters and their interactions, it is clear that the geometries and dimensions of the microchannels will play a critical role in enabling capillary microfluidic devices to function as intended. Some examples of tasks that can be performed with capillary microfluidic designs include capillary pumps [10,11] and capillary valves [12], for the control of the capillary flow velocity within microchannels. Together, these individual functions can be incorporated into a single microfluidic device to carry out a complex flow regime to fulfill specific research goals [13,14,15]. A new capillary microfluidic valve design allowed fluid flow to be turned off and on in a sequential manner to time the delivery of chemicals in the incubation period, which is a common feature of immunoassay experiments [13]. A capillary microfluidic device has also been designed for automated operation. By designing multiple microfluidic components in parallel, the complete unit can perform chain reactions that involve the release of hundreds of aliquots in sequence, the detection severe acute respiratory syndrome-coronavirus-2 (SARS-CoV-2) antibodies in saliva samples, and the continuous sampling of coagulation-activated plasma as part of an assay for thrombin generation [14]. Another example is a one-step immunoassay capillary microfluidic device that integrated a series of capillary triggers, critical retention valves, and retention burst valves to allow users to perform the complete sequence of immunoassay steps, simply by dropping the required reagents into the device [15].

To enable capillary microfluidic devices to perform advanced functions, highly specific microchannel dimensions and geometries must be met during fabrication. This includes microfluidic designs that go beyond conventional single-layer microchannel circuitry and incorporate multilayer microchannel designs with microchannels of different aspect ratios [15]. The fabrication of multilayer microfluidic devices has used conventional photolithography [15] as well as alternative methods, such as micro-milling [11,13,16,17] and resin 3-dimensional (3D) printing [14]. However, direct laser fabrication methods, although widely used for microfluidic fabrication owing to their cost-effectiveness and ease of operation [18,19,20,21,22], have not been extensively explored for use in multilayer microfluidic fabrication.

In our previous publication, we reported a fabrication protocol for single-layer microfluidics on a flexible polydimethylsiloxane (PDMS) substrate using a diode laser machine and Kapton tape [22]. A commercial diode laser Computer Numerical Control (CNC) engraver and Kapton tape were used to fabricate basic microfluidic chips using a maskless, single-step fabrication process under normal atmospheric conditions. In this study, we extended our previous efforts by developing a fabrication protocol for multilayer microfluidics on solid polymethyl methacrylate (PMMA) substrates. This was achieved by repeating laser ablation on the same path to create microfluidic features of varying depths and widths in a single microfluidic unit. The main objective was to create microfluidic designs with highly customizable microchannel dimensions, which is a requirement for more functional and complex fluid control in capillary microfluidic systems.

## 2. Materials and Methods

The fabrication protocol in this study is based on our previous work on direct microfluidic fabrication on PDMS substrates, designed to be highly accessible regardless of the technical expertise available or budgetary constraints [22]. The primary instrument and material used in that work as well as basic fabrication protocols and settings, were retained in this study, with further modifications presented here. Microfluidic designs were created as 2-dimensional sketches using Autodesk Fusion 360 computer-aided design (CAD) software (Version 2.0.15050, Mill Valley, CA, USA), and were exported as dxf files for the fabrication process. The primary instrument for microfluidic fabrication was a Snapmaker 2.0 Modular 3-in-1 3D printer A250 with a diode laser module of 1600 mW output at 450 nm wavelength (Shenzhen, China). The CAD files in dxf format were then imported to Snapmaker bundle software Luban (Version 4.4, Shenzhen, China), where all laser control and operation can be performed.

All microfluidic fabrications were performed on clear polymethyl methacrylate (PMMA) sheets, approximately 1 mm thick. To prepare the substrate for fabrication, Kapton tape, 3M™ polyimide film tape 5413 (Saint Paul, MN, USA), was applied to the top surface of the substrate, covering a surface area great enough for the microfluidic design to be fabricated on the tape. A digital vernier caliper was used to measure the combined material thickness, which was inputted into the software to operate the auto-focus function. The PMMA substrate with Kapton tape was then placed on the workspace of the Snapmaker, ready for laser fabrication. After laser ablation, the remaining polyimide film tape was removed, and the fabricated microfluidic product was cleaned with isopropyl alcohol (IPA) to remove any residual adhesive and carbon particles.

### 2.1. Fabrication Modes: Raster vs. Vector

Two fabrication modes were used in this study: raster mode and vector mode. While in raster mode, the microfluidic designs are read in software as raster files (jpg, png, etc.) and as solid 2D shapes filled with individual dots at specified intervals for laser ablation. The laser ablation process follows a user-defined protocol where various parameters are set. Key parameters include the fill interval, duration of laser ablation time on individual dots, and the laser power applied. For the highest possible quality of finished product, the fill interval is set at 0.05 mm, which is the lowest setting allowed by the software. Laser ablation time was set at 21.5 milliseconds per dot.

In vector mode, the microfluidic designs are read in software as vector drawings, and the laser ablation path follows the lines that appear in the vector image. Vector mode fabrication directly creates microchannels along the path of laser ablation, based on software parameters set by the user.

The laser power used in this study ranged from 30%/480 mW, 40%/640 mW, and 50%/800 mW to 60%/960mW, and the number of laser ablation repetitions varied from 1 to 5. The laser movement speed was set at 140 mm/min.

### 2.2. Microchannel Cross-Section Profile Analysis

Microchannel cross-section profile analysis was designed to determine the microchannel dimensions, specifically the depth, width, and cross-section profile of the microchannels. Straight microchannels were created on PMMA substrates using both raster and vector modes, with different laser power settings and number of repetitions. The microchannel widths tested were 0.5 mm and single line, which was the narrowest width made possible by the Luban software. Examples of laser ablation paths for raster mode and vector mode for single-line and 0.5 mm microchannels created with Luban software are shown in Figure 1.

A Zeiss Microscope with an Axiocam 105 color digital microscope camera with Zeiss Zen 2.3 Lite software (Jena, Germany) was used to take digital photographs of microchannel cross-section profiles. Microchannel dimensions were measured using ImageJ software (Version 1.53q).

### 2.3. Geometry Fabrication Tests

Geometry fabrication tests were performed to determine the suitability of the fabrication protocol for producing geometric components of microfluidic designs. These geometric components, such as circular and rectangular shapes, are integrated within microfluidic designs and function as reservoirs, inlets, outlets, flow resistance or reaction, or detection chambers. Two different paths were tested in vector mode for fabrication of solid geometries. The first path was horizontal, starting from the top left corner of the drawing and moving down, line by line, until completion. The second path was to follow the geometric outline of the drawing, starting from the outside and moving inward to fill the area.

Examples of laser ablation paths of all modes for circular shapes, as created in Luban software, are shown in Figure 2.

### 2.4. Line Fabrication Tests

To test the quality of microfluidic channel segments, sets of lines were created using both vector and raster modes. The lines tested included straight, curved, and intersecting lines, and they were fabricated with the narrowest setting permitted by the software. The objective was to test whether continuous, unbroken lines could be created with the proposed fabrication methods, in order to ensure uninterrupted fluid flow during operation. CAD drawings of all the lines tested in this study are shown in Figure 3.

### 2.5. Multilayer Microfluidic Fabrication Procedure

To fabricate a multilayer microfluidic design on PMMA substrate, the procedure is similar to single-layer microfluidic design with slight modifications. Initial setup of the substrate and polyimide tape for laser auto-focus needed to be performed only once. Then, individual laser protocols for different layers were setup sequentially in Luban software, starting with the deepest layer. The laser ablation process carries out all protocols in order, as setup in the software to produce the final microfluidic device.

### 2.6. Capillary Microfluidic Flow Test

Before conducting a capillary microfluidic flow test, the microfluidic device has to be treated with plasma to render its surface hydrophilic. Plasma cleaner, Harrick Plasma PDC-32G-2 (NY, USA), was used to clean the microfluidic surface. The microfluidic device was placed inside the chamber and vacuum-sealed, before activating the plasma at a high-power setting (18 watts) for 1 min. The device was then ready for use, and fluid was introduced manually with a syringe or pipette.

## 3. Results

### 3.1. Microchannel Cross-Section Depths and Widths

A complete set of graphical data (*n* = 3) of the microchannel dimensions is shown in Figure 4. As expected, when higher laser power levels were used for microchannel fabrication, the resultant microchannels were wider and deeper. This was because more laser thermal energy was absorbed by the polyimide tapes and transferred to the PMMA substrates, resulting in more laser ablation, and creating deeper and wider microchannels. Following the same trends, increasing the number of laser ablations during fabrication also produced deeper and wider microchannels in general. However, this effect of repeated laser ablation on microchannel dimensions did not prove to be true for all the protocols tested. Using the raster 0.5 mm protocol (Figure 4c), the microchannel width and depth increased as a higher laser power was applied. However, with increasing laser repetition at a constant laser power, there was no significant increase in the microchannel width, while the depth increased accordingly. A similar trend was also observed with the raster single-line protocol (Figure 1a), with one significant difference. The microchannel width after a single laser ablation was significantly lower than that created by laser repetitions, which showed no significant differences across the entire number of laser repetitions.

A closer look at the data sets on the fabrication protocols showed different cross-sectional dimensions between the raster and vector fabrications. Microchannel fabrication with vector protocols resulted in a greater width across all laser powers and the number of repetitions, compared to the microchannel width from raster fabrication. One possible explanation is that the vector protocol delivered a continuous laser output along the whole length of the microchannel, unlike the raster protocol, in which individual dots along the line received laser energy for a fixed amount of time before the laser module moved on to the next dot location in the line, so that the laser energy was delivered in a pulsed manner with frequent pauses. Although the movement speed in the vector mode and the laser ablation time per dot in the raster mode were calculated to match each other, the results showed that the vector mode produced wider microchannels, due to the continuous and uninterrupted laser energy delivery to the substrates.

The data showed that it is possible to create microchannels with specific widths and depths by adjusting the laser power and the number of laser repetitions while using this protocol. A quick example of how to achieve a specified microchannel dimension is to first select the laser power to produce the required microchannel width, then select the number of laser repetitions to produce the required microchannel depth. While there are some limitations to the range of microchannel dimensions that can be fabricated, this protocol allows users to easily create microchannels of specific widths and depths for their research and applications. This property will become highly desirable within the context of passively driven microfluidic devices, such as capillary microfluidics, where the dimensions and cross-section aspect ratio of the microchannel are the critical factors that govern the fluid flow within the microfluidic system.

### 3.2. Cross-Section Profile Analysis

Figure 5 shows examples of cross-section profiles of the microchannel fabricated using different modes with 40% laser power and up to five repetitions. For a complete set of examples fabricated with other laser power settings, please refer to the Appendix A. The microchannel cross-section dimensions for all protocols tested follow a similar general trend. As the laser power, number of laser repetitions, or the combination of both increased, the microchannel depths and widths become larger, in agreement with the numerical data presented in the section above. Actual dimensions aside, there are other features of the microchannel profiles that are noteworthy and may be relevant to microfluidic flow and operation.

Fabrication with the vector single-line and raster single-line protocols showed similar cross-section profiles of the resultant microchannels. The profiles of the microchannels fabricated with one and two laser repetitions appeared to be C-shaped, with a soft curvature and no clear straight line along the microchannel wall. As the number of repetitions increased from three to five, the cross-section profile started to become V-shaped, with a pointed bottom and almost straight lines on the walls on both sides.

Microchannels fabricated with the 0.5 mm vector protocol had highly irregular, non-uniform cross-section profiles, with no clear trend or any distinct features, across all the laser power settings and repetition numbers.

The fabrication of microchannels with the 0.5 mm raster protocol showed the highest quality and the most consistent surfaces and cross-section geometry with all the laser power settings. With the different numbers of laser repetitions, the shapes of the channel’s cross-sections changed. With a single laser ablation, the cross-section profile had a wide C-shape. As the number of laser repetitions increased, the channel geometry started to take on a trapezoidal shape, with the wider side at the top and the narrow side at the bottom of the channel. However, with four and five laser repetitions, there was a significant degradation of the channel surface, specifically the bottom, where the flat-level surface showed pronounced depressions at the corners.

In general, the raster fabrication produced more consistent and uniform rectangular cross-section profiles than the vector fabrication with the same settings. Single-line fabrication from both raster and vector modes produce V-shaped profiles with three or more laser repetitions. The optimal setting for the rectangular microchannel profile is fabrication in the raster mode at 0.5 mm with three laser repetitions. Note that rectangular geometry is the most common geometry for capillary microfluidic devices, largely due to the use of photolithography or alternative, rapid fabrication methods [23]. This information may be relevant when attempting to recreate microfluidic designs and flow regimes and for numerical simulations or modeling purposes, which will most likely use rectangular microchannels. More than three laser repetitions very quickly degrade the cross-section profiles and wall surfaces, and their use is not recommended.

### 3.3. Geometry Fabrication Tests

Geometry is the term used to differentiate other microfluidic features from microchannels that usually appear as thin lines. Common microfluidic features include microchannels, inlets, outlets, reservoirs for various chemicals and reagents, incubation chambers, and electrochemical cells for microfluidic device with integrated sensor units. The most common shapes used in microfluidic devices are circles and rectangles, both of which were selected for the geometry fabrication tests in this study. Figure 6 shows examples of geometries fabricated with the raster and vector modes using laser powers 30%, 40%, 50%, and 60% and 3x repetitions. In each sample photograph, the left column shows rectangular geometries, and the right column shows circular geometries. The top row shows geometries made in the raster mode, while the middle row and bottom row show geometries made in the vector mode path one and path two, respectively. For geometric shapes, raster fabrication produced more consistent and level geometric shapes and outlines, with noticeably sharp and clean transitions between different depths. The texture of the ablated surface also appears to show consistent, non-uniform surface roughness. In contrast, vector fabrication produced a laser ablation path visibly engraved into the substrate, with path one showing the outline of the geometry and path two showing horizontal lines. Additionally, the shape outline produced is not as clean or consistent as with raster fabrication. These effects became more pronounced with increased laser power during the fabrication process.

### 3.4. Line Fabrication Tests

A series of lines fabricated in these tests represents the microchannels that appear in microfluidic devices. The purpose of this test was to determine whether the fabrication protocol can produce the accurate and continuous lines that are essential for an uninterrupted fluid flow. Figure 7 shows the results of line fabrication tests performed using the raster and vector modes to fabricate the smallest possible microchannel width on PMMA substrate using a laser power of 30% and three repetitions.

A single-line raster fabrication did not consistently produce complete lines, especially diagonal or curved lines. This was due to the smallest interval setting in the Luban software (0.05 mm) not being close enough, creating gaps between dots that were not located on the 90-degree vertical or horizontal axis. This was most likely due to the spacing of the dots on the diagonal being greater than those on the vertical and horizontal axes. Additionally, laser ablation did not remove enough material from the substrate to connect the dots and form a continuous microchannel. This issue may be circumvented by using a higher laser power to produce larger individual dots that are connected to the neighboring dots, forming a continuous channel. However, the overall microchannel width and depth will then be larger, which may not suit all applications.

A possible cause of this issue with the raster fabrication of narrow microchannels was how the Luban software created a G-code for laser fabrication based on the design. With the microchannel width in the CAD software set at 0.025 mm, the Luban software interpreted the data and wrote the subsequent G-code differently. In certain cases, based on the direction of the lines, the same microchannel width can result in a single line of dots with laser ablation, two parallel dotted lines with a greater line width, or no dotted lines at all. Consequently, the resulting microchannel produced on the substrate will vary in size. This issue was improved as the width of the microchannel increased beyond the minimum of 0.05 mm. Since this issue seems to be hardware- and software-specific, one possible solution is to use a more complex laser system and control software or implement some manual modifications of the G-code for more direct control of the CNC system. However, this option would render the fabrication method much less cost-efficient and user-friendly.

Based on this set of data, only the vector method is recommended for the fabrication of the smallest or narrowest microchannels. For microchannel widths larger than 0.05 mm, both the raster and vector modes should result in microchannels with functional fluid flow.

### 3.5. Capillary Flow Test–Trigger Valve

To confirm that the fabrication protocol presented here can produce functional capillary microfluidic devices, a multilayer microfluidic design with a trigger valve was created. The trigger valve microfluidic unit consisted of two microchannels, a main channel with a serpentine section and circular reservoirs at both ends, and a smaller side channel that joined the main channel. For the fabrication process, the main channel width was set at 0.3 mm and was created using 50% laser power with three repetitions. The side channel width was set at 0.1 mm and created using 30% laser power with three repetitions. Both microchannels were fabricated using the raster protocol for the optimal fluid flow, due to the wide, rectangular cross-section profile. Diagrams of the trigger valve and photographs of the sequence of the fluid flow are shown in Figure 8. The sequence of the fluid flow shows photographs of the trigger valve before use, with the side channel loaded with red dye and the main channel loaded with green dye (Figure 8c).

Before operation, the fabricated microfluidic unit with the trigger valve was plasma-cleaned to render the microchannel surface hydrophilic. A common test for surface hydrophilicity is the water contact angle test, and for the purpose of capillary microfluidic flow, a minimum water contact angle of 60° is recommended [23]. When this requirement is met, capillary action is generated by the concave liquid–air interface and negative capillary pressure, causing spontaneous liquid wicking effects along the microchannels. During the operation, red dye was first loaded into the side channel, where it flowed along the path and stopped at the intersection with the main channel, due to the abrupt change in microchannel geometries and the consequent drop in capillary pressure. Based on the Young–Laplace equation within a rectangular microchannel [24], the capillary pressure is inversely correlated with the smallest dimension of the microchannel. This caused the fluid flow in the smaller microchannel to stop spontaneously at the intersection with the larger microchannel, where the capillary pressure dropped. With a multilayer microfluidic trigger valve, such as the one tested here, having two microchannels of different depths, creating a two-tier trigger valve, greatly facilitated the sudden change in capillary pressure necessary to stop the initial flow. A green dye was then added to the main channel and flowed along it. At the junction with the side channel, the trigger valve was activated, causing the fluid in the side channel to join the fluid in the main channel. As both microchannels at the junction were filled with fluid, the requirement for a trigger valve had been met and both fluids continue on their path [12,25]. Fluid from both microchannels continued to flow in the main microchannel until they reached the reservoir at the end of the path. Notice that in the microchannel section after the trigger valve, the green dye from the main channel and the red dye from the side channel started to flow alongside each other, gradually mixing as they progressed along the main channel.

## 4. Conclusions

This study serves as a starter guide for the fabrication of a multilayer microfluidic device that can be carried out quickly, easily, and with minimal investment. The protocol introduced here enables a user to make microfluidic devices using a diode laser CNC machine with polyimide tape on polymer substrates in a regular benchtop setting, bypassing any requirement for conventional microfluidic fabrication, such as the cleanroom environment that is necessary for photolithography. The protocol is arguably the most accessible and inexpensive solution for fabricating microfluidic devices, even when compared to other widely used fabrication methods, such as 3D printing and micro-milling. The laser fabrication protocol detailed here has demonstrated that multilayer microfluidic devices can be created to the required specifications for highly controlled capillary fluid flow, which other direct laser fabrication studies have not demonstrated. The protocol allows users to fine-tune the dimensions of the microchannel to create a specific aspect ratio of the cross-section profiles that best suit their intended goals. This is possible by using the raster fabrication mode, in which a specific microchannel width can be set, and the microchannel depth can be adjusted through the number of laser repetitions. This allows for a greater level of control of the microfluidic flow regime, which is crucial in many applications and systems, particularly passively driven or capillary microfluidics.

The low startup and operating costs, ease of operation, and a fabrication protocol that accurately produces a specific microfluidic channel aspect ratio, are the key advantages of this fabrication method. This allows users to make capillary microfluidic devices very quickly and cheaply, making the technique suitable for rapid prototyping and removing any barrier of entry for any research groups or individuals seeking to integrate microfluidic technology into their research endeavors and product development. Because of the ease of use for the end-users and the lack of requirements for external equipment when compared to conventional microfluidics, capillary microfluidic devices are highly suited to the development of portable, miniaturized devices for real-world uses in locations, including POC devices for medical diagnosis, especially in resource-poor communities where quality healthcare may be scarce.

## Figures and Tables

**Figure 1 micromachines-14-00324-f001:**
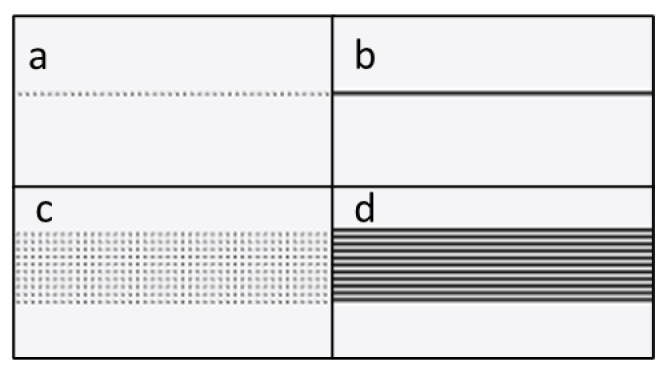
Laser ablation path for different modes, (**a**) raster single-line, (**b**) vector single-line, (**c**) raster 0.5 mm, and (**d**) vector 0.5 mm.

**Figure 2 micromachines-14-00324-f002:**
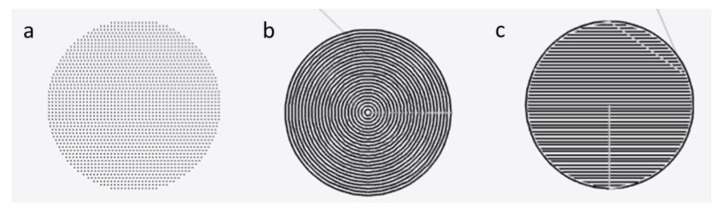
Laser ablation path for sample geometries under different modes, (**a**) raster, (**b**) vector path 1, and (**c**) vector path 2.

**Figure 3 micromachines-14-00324-f003:**
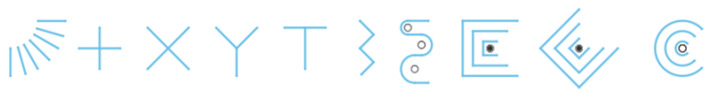
CAD drawings of lines used in fabrication tests.

**Figure 4 micromachines-14-00324-f004:**
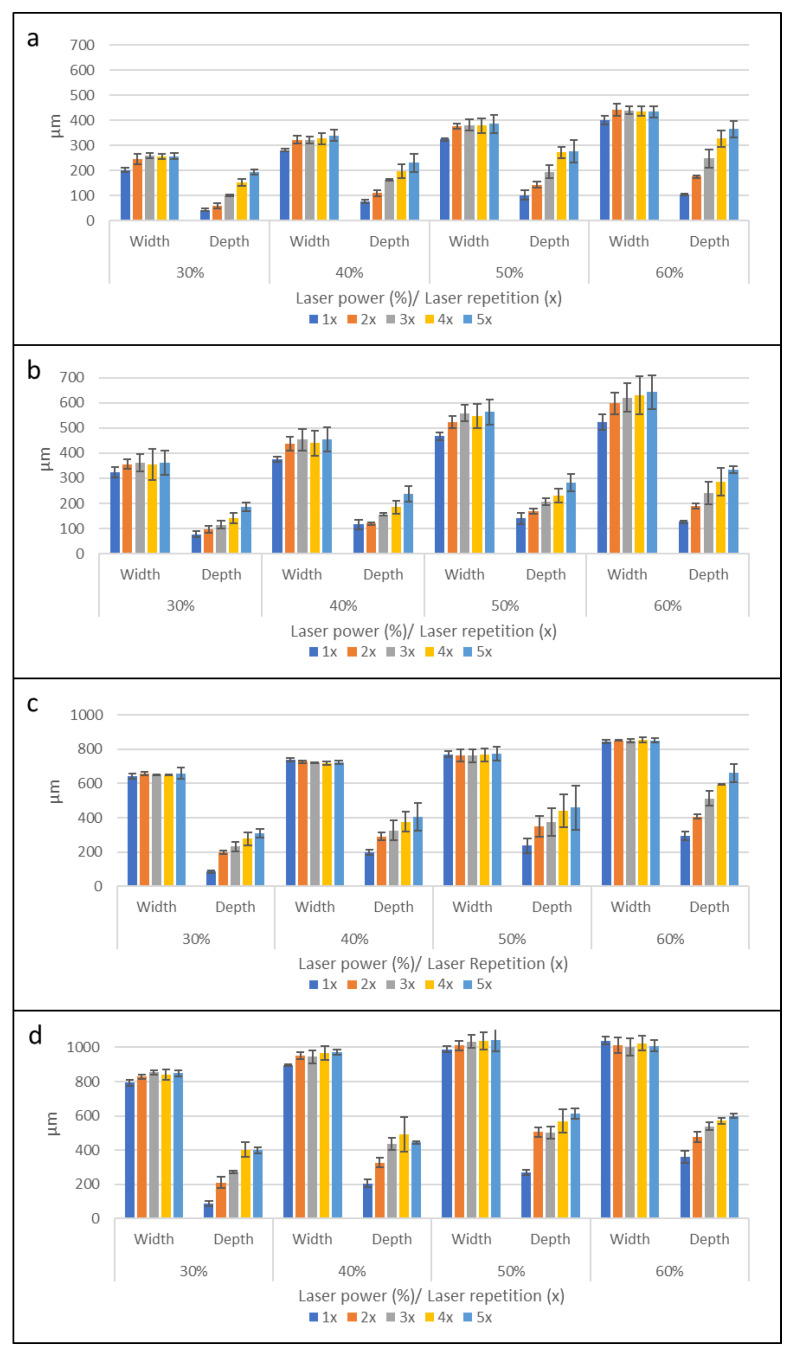
Microchannel cross-sectional widths and depths for single-line and 0.5 mm microchannels fabricated by the raster and vector modes, using the combination of laser power at 30%, 40%, 50%, and 60% with laser repetition numbers 1x, 2x, 3x, 4x, and 5x (*n* = 3). (**a**) Raster single-line, (**b**) vector single-line, (**c**) raster 0.5 mm, and (**d**) vector 0.5 mm.

**Figure 5 micromachines-14-00324-f005:**
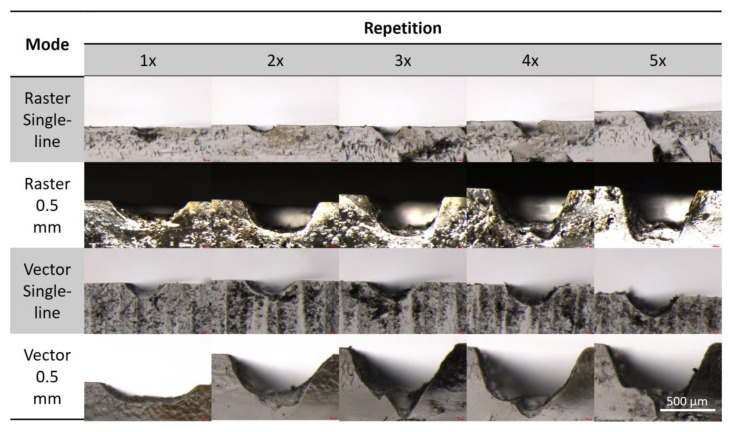
Photographs of cross-sectional profiles of microchannels fabricated with 40% laser sample with up to 5 repetitions on PMMA substrates.

**Figure 6 micromachines-14-00324-f006:**
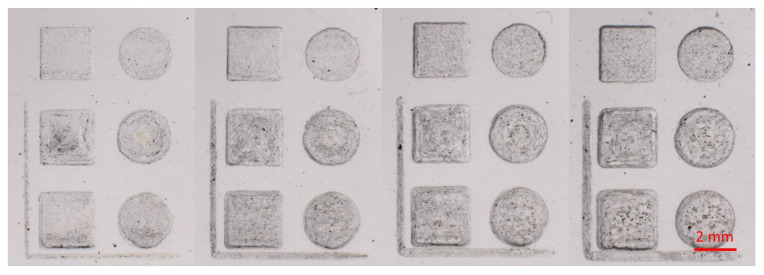
Photographs of geometry fabrication tests using, from left to right, 30%, 40%, 50%, and 60% laser power and 3 repetitions on PMMA substrates.

**Figure 7 micromachines-14-00324-f007:**
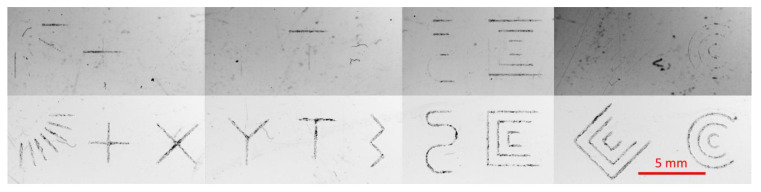
Photographs of sets of lines fabricated using raster mode (top row) and vector mode (bottom row) on PMMA substrates.

**Figure 8 micromachines-14-00324-f008:**
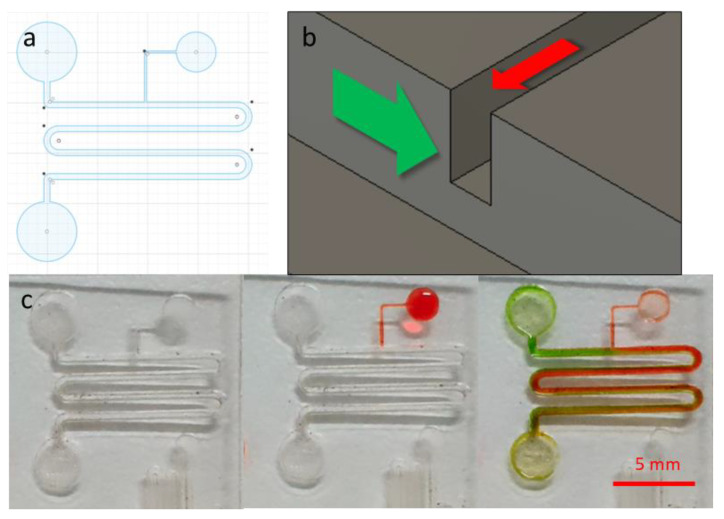
Trigger valve microfluidic design and operations. (**a**) CAD drawing of the microfluidic with trigger valve, (**b**) 3D schematic of fluid colors and flow directions of trigger valve, and (**c**) sequence of photographs of trigger valve operation.

## Data Availability

All data are included in the manuscript.

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
