# Peer review of "Basic Guide to Multilayer Microfluidic Fabrication with Polyimide Tape and Diode Laser"

_micromachines, 2023, doi:10.3390/mi14020324_

Round 1

Reviewer 1 Report

Title:

 Basic Guide to Multilayer Microfluidic Fabrication with Polyi-2 mide Tape and Diode Laser

General Comments:

The paper is of current interest and falls in the journal's scope well. However, the following changes would make it more interesting for the readers:

1. The article needs a thorough check on the spelling and grammar of sentences. Many sentences are unclear to understand for the general audience. Therefore, it is recommended that the authors must take services from a native English speaker.

2. The author has cited several relevant references to build up the present model's literature review. However, it is not sufficient. It is vast but not comprehensively focused on the relevance of the problem chosen in the manuscript. Authors should update the introduction section by including recent articles published explicitly in the last two years.

3. Experimentally, it is good work treated with a numerical procedure. Does it have an equivalent theoretical relevance? Can it be compared with a theoretical approach on the same topic?

4. The methodology section needs further expansion.

5. Clarify and discuss the novelty and significance of the results obtained here, compare them with those available in the literature, and discuss potential applications and if possible, pointwise provide the improvements as compared to the previous model of this paper.

6. Geometry and modelling need further explanation/justification and literature referencing.

7. The results and discussion section is comprehensive. However, it needs more theoretical reasoning behind each development concerning some physical meaning of the scenario.

8. Conclusions should be revised, the current findings are most general, and the authors are requested to highlight the critical investigations of the present study in the conclusion section.

9. Cite suitable references for all the governing equations.

10. Provide a nomenclature table for all the abbreviations and symbols. Add SI units in the nomenclature table (if needed).

11. Some figures are unclear. Specifically, Fig. 6 and 7 need high resolution.

Reviewer 2 Report

In this paper, the authors demonstrate the fabrication protocol for multilayer microfluidics on solid PMMA substrates with Diode laser and polyimide tape. Various microchannel depths and widths can be created by adjusting the used laser power and the number of laser repetitions under suitable modes. This protocol shows the potential to create microfluidic channels with highly customizable microchannel dimensions, which is a requirement for more functional and complex fluid control in a capillary microfluidic system. The logic of this manuscript is clear, and the experimental design is reasonable. However, there are still some problems that need to be addressed.

1. Since the fabrication process involved multiple layers, the authors need to care about the change of laser focus on the substrate interface.

2. In this manuscript, the authors investigated the influence of laser power, repetition times, and fabrication modes on the preparation. However, laser movement speed, Pixels Per Inch (PPI) and Dots Per Inch (DPI) values will also affect the geometry and the quality of the device. The authors should give a comprehensive investigation of these parameters.

3. For Figure 4, the authors need to give the number of samples for each group of data. In addition, it would be better to adjust the column or err bar width to make the figure clear.

4. The authors need to give the size bar of figure 8c.

Reviewer 3 Report

Basic Guide to Multilayer Microfluidic Fabrication with Polyimide Tape and Diode Laser

As the title suggested, the paper was a guide for fabricating PMMA devices with polyamide tapes. The fabrication parameters were studied and well-documented. However, some 

aspects of the paper can be improved.

First, the paper's motivation is unclear - the authors should better (in more detail) address the unique benefit of using these techniques. Is it for chemical reaction or cell culture? What is the downstream application that this fabrication is best for? What is the advantage?

More specific questions:

Figure 2: At least four light gray lines in the second and third vector path circles. What are they? Why does it even extend beyond the circle path? Please also label 1.2.3 in the figure. 

Figure 3. What is the difference between black and blue lines? 

Figure 4. what does the error bar represent? 

Round 2

Reviewer 1 Report

I have no more concerns. 

Reviewer 2 Report

The authors have addressed all the reviewer's comments completely. In this regard, I recommend the acceptance of this manuscript.